# Entomopathogenic Bacteria Species and Toxins Targeting Aphids (Hemiptera: Aphididae): A Review

**DOI:** 10.3390/plants14060943

**Published:** 2025-03-17

**Authors:** Erubiel Toledo-Hernández, Mary Carmen Torres-Quíntero, Ilse Mancilla-Dorantes, César Sotelo-Leyva, Edgar Jesús Delgado-Núñez, Víctor Manuel Hernández-Velázquez, Emmanuel Dunstand-Guzmán, David Osvaldo Salinas-Sánchez, Guadalupe Peña-Chora

**Affiliations:** 1Facultad de Ciencias Químico Biológicas, Universidad Autónoma Guerrero, Av. Lázaro Cárdenas s/n., Chilpancingo C.P. 39070, Guerrero, Mexico; madoil@icloud.com (I.M.-D.); cesarsotelo@uagro.mx (C.S.-L.); 2Centro de Ciencias Genómicas, Universidad Nacional Autónoma de México, Av. Universidad #1001, Col. Chamilpa, Cuernavaca C.P. 62209, Morelos, Mexico; mark_tq@hotmail.com; 3Facultad de Ciencias Agropecuarias y Ambientales, Universidad Autónoma Guerrero, Iguala de la Independencia C.P. 40020, Guerrero, Mexico; edgarjezus@gmail.com; 4Centro de Investigación en Biotecnología, Universidad Autónoma del Estado de Morelos, Av. Universidad #1001, Col. Chamilpa, Cuernavaca C.P. 62209, Morelos, Mexico; vmanuelh@uaem.mx; 5Facultad de Ciencias Agropecuarias, Universidad Autónoma del Estado de Morelos, Av. Universidad #1001, Col. Chamilpa, Cuernavaca C.P. 62209, Morelos, Mexico; dunstand_ipalogy@outlook.com; 6Centro de Investigación en Biodiversidad y Conservación, Universidad Autónoma del Estado de Morelos, Av. Universidad #1001, Col. Chamilpa, Cuernavaca C.P. 62209, Morelos, Mexico; davidos@uaem.mx; 7Centro de Investigaciones Biológicas, Universidad Autónoma del Estado de Morelos, Av. Universidad #1001, Col. Chamilpa, Cuernavaca C.P. 62209, Morelos, Mexico

**Keywords:** bacterial metabolites, biosurfactants, entomopathogens, pest, biological control

## Abstract

Aphids (Hemiptera: Aphididae) are cosmopolitan generalist pests of many agricultural crops. Their ability to reproduce rapidly through parthenogenesis allows them to quickly reach population sizes that are difficult to control. Their damage potential is further exacerbated when they act as vectors for plant pathogens, causing diseases in plants. Aphids are typically managed through the widespread use of insecticides, increasing the likelihood of short-term insecticide resistance. However, for the past few decades, entomopathogenic bacteria have been used as an alternative management strategy. Entomopathogenic bacteria have demonstrated their effectiveness for biologically suppressing insect pests, including aphids. In addition to identifying bacterial species that are pathogenic to aphids, research has been conducted on toxins such as Cry, Cyt, Vip, recombinant proteins, and other secondary metabolites with insecticidal activity. Most studies on aphids have been conducted in vitro, exposing them to an artificial diet contaminated with entomopathogenic bacteria or bacterial metabolites for periods ranging from 24 to 96 h. The discovery of new bacterial species with insecticidal potential, as well as the possibility of biotechnological applications through the genetic improvement of crops, will provide more alternatives for managing these agricultural pests in the future. This will also help address challenges related to field application.

## 1. Introduction

The family Aphididae (Hemiptera) currently consists of at least 4700 known species worldwide; of these, around 100 species have been identified as agricultural pests. Their potential as pests is due largely to their capacity for rapid reproduction through parthenogenesis [1]. They often do not need to reproduce sexually, especially under warm-weather conditions. Some aphids can produce up to 96 nymphs per female, in addition to spreading rapidly following the production of winged forms [2,3]. Such rapid reproduction can lead to enormous aphid populations and extensive crop damage. For example, between 2013 and 2015, the destructive capacity of the sugarcane aphid, *Melanaphis sorghi* (Theobald), was demonstrated by billions of dollars of damage to the sorghum crops of the United States and Mexico [4,5]. In some cases, the crop losses are exacerbated when aphids transmit plant pathogens, particularly viruses. Pesticides are often used to keep aphid pests under control, increasing the probability that some individuals will develop insecticide resistance and pass this characteristic to new generations of aphids [6,7].

Entomopathogenic bacteria have always played a very important role as biological control agents of insect pests, including the regulation of insect populations through natural epizootics [8,9]. Various bacterial families are known to harbor entomopathogenic species that target insect orders such as Lepidoptera, Coleoptera, Hemiptera, Diptera, Orthoptera, Hymenoptera, and Mallophaga, as well as other invertebrate pests from the classes Rhabditida, Gastropoda, and Arachnid [10,11,12]. Among these, the families Bacillaceae, Pseudomonadaceae, Enterobacteriaceae, Lactobacillaceae, Planococcaceae, Micrococcaceae, Paenibacillaceae, Streptomycetaceae, and Neisseriaceae include species that are pathogenic to insects [13,14,15,16,17]. Several studies have shown that within these families of bacteria, there are species that are pathogenic to aphids, and some strains have already been formulated and patented in the United States as bioinsecticides against these insects [13,18,19]. In addition to the search for and molecular identification of aphid pathogenic strains, great emphasis has been placed on trying to identify the bacterial pesticidal proteins (BPPs) that could be responsible for aphid toxicity. BPPs include Cry, Cyt, vegetative insecticidal proteins (VIPs), and other molecules such as biosurfactants. Also, studies have been carried out to explain the possible mechanism of action by which some toxins are able to cause death in this group of insects [20,21,22]. However, the mechanism of action of BPPs in aphids has not been fully resolved.

On the other hand, only a limited number of BPPs or other secondary metabolites with toxicity against aphids have been reported. The low efficacy of many BPPs and secondary metabolites against these hemipterans is related to various factors, including their specialized feeding behaviors and structural and chemical aspects of the gut that decrease exposure, such as the presence of a filter chamber, the absence of a peritrophic membrane, and suboptimal conditions for BPP activation, processing, and/or binding [23,24].

Although bacteria account for most of the market for biological insecticides, there are no products specifically offered for aphids. Although aphids have low susceptibility to original versions of the Cry and Cyt toxins of *Bacillus thuringiensis* (Bt), the toxicity of these molecules can be significantly increased by the addition of peptides through protein engineering [23]. Likewise, the initial solubilization and proteolytic activation of the toxin increases its biocidal effect [9,25,26]. Although it has been identified that aphids lack some genes for the immune response to pathogens, the role played by aphid endosymbiont organisms is not minor. Several studies have shown that aphid endosymbionts protect their hosts from pathogens, parasitoids, and stress [27]. The analysis of genes related to stress and immunity in this group of insects will help in reconstructing the evolutionary history of innate immunity and other defenses against entomopathogens.

Currently, the field application of entomopathogenic bacteria or toxins against aphid pests presents challenges and limitations. Most studies of entomopathogenic bacteria against aphids have been conducted in vitro, exposing the aphid for 24 to 96 h to an artificial diet contaminated with the entomopathogenic bacteria or its toxins. These exposure conditions can be challenging to reproduce in the field. Efforts to find bacteria with the potential for application in the field for the control of sap-sucking pests are needed, including those with a biocidal effect on aphids after a short exposure time to the bacteria.

The objective of this review is to explore the potential of entomopathogenic bacteria and bacterial pesticidal proteins (BPPs) for the biological control of aphid pests, with a focus on understanding their mechanisms of action, enhancing their efficacy through protein engineering, and addressing challenges related to field application.

## 2. Entomopathogenic Bacterial Families That Affect Aphids

### 2.1. Bacillaceae

The family Bacillaceae contains some of the most resilient bacteria on Earth; this is mainly due to their ability to form resistant endospores, which are a key factor determining the ecology of these bacteria. Bacillaceae is widely distributed in natural environments, and their habitats are highly varied. They are rod-shaped and ubiquitous, and the majority of this group are facultatively aerobic or anaerobic and chemoorganotrophic [28]. Furthermore, they play essential roles in soil ecology in the cycling of organic matter and in stimulating plant growth (e.g., through the suppression of plant pathogens and phosphate solubilization) [29]. This family contains species that are effective as biological control agents of agricultural pests and disease vectors [30].

Within the *Bacillus* genus of the Bacillaceae, the species *Bacillus thuringiensis* is considered the most important bacterium for pest control in both agricultural crops and in medical contexts [31]. The effectiveness of this bacterium is attributed principally to the production of two groups of toxins known as Cyt and Cry delta-endotoxins [31]. These toxins have been extensively studied by various research groups and have demonstrated insecticidal activity against different orders of insects, including the order Hemiptera [32]. The earliest work in the search for aphid entomopathogens was reported by Payne and Cannon [18], who patented three strains of *B. thuringiensis* as pathogens against aphids (patent: US 5262159 A). However, no commercial products based on these strains are currently available for use against Hemiptera. Hence, different research groups have evaluated different bacterial isolates in an attempt to find a strain with the potential to control this group of insects. For example, Malik and Riazuddin [33] identified three strains (INS 2.13, HFZ24.8, GU 9.1) with insecticidal activity against the aphid *Aphis gossypii* (Glover), with a median lethal concentration (LC_50_) of 62 ng/mL for strain INS 2.13, 328 ng/mL for strain HFZ24.8, and 114 ng/mL for strain GU 9.1, suggesting that these strains produce new toxins that may be of great importance for the control of this aphid species.

On the other hand, Monnerat and colleagues [34] evaluated 400 strains of *B. thuringiensis* against *A. gossypii.* The methodology of the pathogenicity bioassay involved placing a cotton plant leaf in contact with a solution contaminated with the strain to be evaluated. Out of the 400 strains assessed, only five strains were pathogens of the aphid, causing mortalities greater than 50%. In another study, Alahyane and colleagues [35] evaluated the insecticidal effect of 82 *B. thuringiensis* strains from Moroccan crops against *A. gossypii*. They used feeding assays with a feeding membrane and artificial diet to assess the toxicity. Their findings showed that 18.29% of the studied strains were highly toxic against the first and third instar nymphs of the aphid. Among the selected strains, BtA4, BtA1, and Bt21.6 displayed the highest toxicity against the first instar, with LC_50_ values of 0.15 mg/mL, 0.23 mg/mL, and 0.25 mg/mL respectively. Meanwhile, strains BtB6, BtA10, and Bt21.6 showed high nymphicidal potency against the third instar, with LC_50_ values of 0.48 mg/mL, 0.79 mg/mL, and 1.14 mg/mL, respectively.

Rajashekhar and Kalia [36] isolated strains of *B. thuringiensis* from soil samples collected from different locations and evaluated them against *A. gossypii*, evaluating the effect of the protein in three different states: pre-solubilized, solubilized, and trypsinized. The three evaluated strains exhibited mortality rates ranging from 23.33% to 46.66% in the pre-solubilized protein treatment, 30% to 53.3% in the solubilized protein treatment, and 13.33% to 36.66% with trypsinized protein. On the other hand, Ramasamy and colleagues [37] isolated 65 Bacillus-like strains from soil samples and evaluated their insecticidal effect against *A. gossypii* and *Aphis punicae* (Passerini). They used a feeding membrane system with a suspension of spore-crystals in water. Out of these 65 strains, 15 were identified as putative aphicidal *B. thuringiensis*, and only three displayed insecticidal activity against both aphid species, with an LC_50_ of 35 μg/mL for the three strains. The detection of the cry gene of these isolates was carried out by PCR analysis, which indicated that *cry1*, *cry2A*, *cry3A*, and *cry11A* were on plasmids.

Similarly, Torres and colleagues [38] conducted a pathogenicity screening with approximately 40 bacterial strains of *B. thuringiensis* isolated from insect corpses of different families in the order Hemiptera and evaluated them against *Myzus persicae* (Sulzer) as a spore–crystal mix in an artificial diet. Out of the 40 strains assessed, only 17 strains were found to be pathogens of the aphid. At a concentration of 10 ng/μL of the total protein, aphid mortality was 64–88% after three days. López and colleagues [39] evaluated the aphicidal effect of the *B. amyloliquefaciens* strains CBMDDrag3, PGPBacCA2, and CBMDLO3, as well as their metabolites, against *M. persicae*. They tested cell suspensions, heat-killed cell suspensions, cell-free supernatants, and isolated lipopeptide fractions. The different cell fractions were individually offered to aphids through artificial diets, resulting in 100% mortality among adults and nymphs four days after administration. In contrast, the lipopeptide fractions, mainly composed of kurstakins, surfactins, iturins, and fengycins, did not exhibit an aphicidal effect.

On the other hand, some strains of *B. thuringiensis* have been shown to produce VIP secretory proteins during their vegetative growth phase, and the effects of these toxins have started to be evaluated against aphid. Sattar and colleagues [40] conducted a search for VIPs from isolates of *B. thuringiensis*, identifying five strains with insecticidal activity against *A. gossypii* whose mortalities ranged from 30 to 70% after 48 h of exposure. The same authors (2011) later reported new VIPs actives against the same aphid, with mortalities of 50% using 0.576 ng/μL. Additionally, it was recently reported that the genus *Lysinibacillus* also causes mortality in aphids. Baazeem and colleagues [41] report two strains of *Lysinibacillus xylanilyticus* (TU-2 and BN-13) with insecticidal activity against the aphids *Aphis illinoisensis* (Shimer) and *A. punicae*. They evaluated the strains using a cell culture and a concentrated filtrate of the cell culture, which were topically applied to the aphid’s body. They report for strain TU-2 (cell culture) an LC_50_ of 7.41 × 10^4^ CFU mL^−1^ against *A. illinoisensis* and an LC_50_ of 2.51 × 10^4^ CFU mL^−1^ against *A. punicae*. On the other hand, for strain BN-13, they report an LC_50_ of 1.55 × 10^9^ CFU mL^−1^ against *A. punicae* and an LC_50_ of 4.37 × 10^9^ CFU mL^−1^ against *A. illinoisensis*. However, when they applied the concentrated filtrate of the cell culture, they needed doses above 400 μLmL^−1^ to observe 50% mortality in both aphid species.

It is worth noting that the variability observed in the mortality percentages against aphids in the aforementioned reports may be due to variations in the processes of the purification and administration of the toxin-spore-bacteria complex, as well as in the methodology of the different feeding systems used to supply the strain or the spore–crystal complex. On the other hand, there are research groups that have taken on the task of evaluating purified BPPs with the aim of understanding which toxins or components could be exerting an aphicidal effect. Table 1 shows some studies that have evaluated purified BPPs, such as Cry, Cyt, or VIP proteins from *Bacillus* sp., in order to determine if any of them could cause mortality in different aphids, such as *A. gossypii*, *M. persicae*, *Acyrthosiphon pisum* (Harris), *Brevicoryne brassicae* (Linnaeus), and *Macrosiphum euphorbiae* (Thomas). The majority of these studies show low levels of toxicity to aphids even at high doses. Nonetheless, these studies demonstrate that despite their low toxicity, it is highly important to characterize and evaluate the aphicidal effects of these toxins in order to develop an efficient strategy for controlling this group of insects that will be agriculturally significant in the future.

### 2.2. Enterobacteriaceae

The Enterobacteriaceae are a large family of Gram-negative, non-spore-forming bacteria; they are ubiquitous and are distributed across diverse ecological niches in both terrestrial and aquatic environments [47,48]. Although many species are part of the natural microbiome of animals, including humans, some of them are frequently associated with both intestinal and extraintestinal diseases [49]. Furthermore, they can be pathogens against different groups of insects, including aphids. For example, in research by Harada and Ishikawa [50], the pathogenicity of five strains that were isolated from the gut of multiple specimens of the pea aphid *Acyrthosiphon pisum* caused up to 80% mortality in this species when they were applied at a concentration of 10^5^ CFU mL^−1^ in an artificial diet. Furthermore, the results of biochemical tests showed that these strains are related to *Erwinia herbicola* and *Pantoea agglomerans*.

For example, Hashimoto [51], reported that *P. agglomerans* has aphicidal activity against the foxglove aphid *Aulacorthum solani* (Kaltenbach) and the cotton aphid *A. gossypii*. The mortality was evaluated by topical exposure by spraying the bacterial culture and by ingestion using an artificial diet, resulting in nearly 100% mortality five days after inoculation. Similarly, Paliwal and colleagues [52] assessed the toxicity of *P. agglomerans* and *Pantoea* sp. PaR8 against six aphid species (*M. persicae*, *B. brassicae*, *Aphis fabae* (Scopoli), *Macrosiphum albifrons* (Essig), *Nasonovia ribisnigri* (Mosley), and *A. solani*), showing that both strains were pathogenic to all aphid species, resulting in mortality rates ranging from 50% to 70% after 72 h, using bacterial concentrations ranging from 10^2^ to 10^7^ CFU mL^−1^. Conversely, Stavrinides and colleagues [53] identified the bacterium *P. stewartii* ssp. *Stewartii*, a phytopathogenic bacterium of maize plants, as a pathogen of *A. pisum*. Their research revealed that its entomopathogenic potential is attributed to a transmembrane protein named ucp1. Deleting the gene, as well as cloning and expressing it in *E. coli* BL21, confirmed its role in the bacterium’s virulence. The protein enables bacterial aggregation, leading to gut obstruction and subsequent mortality in insects. On the other hand, Campillo and colleagues [54] reported two new species of the genus *Erwinia* (*E. iniecta* B120T and *E. iniecta* B137) isolated from crushed corpses of the Russian wheat aphid, *Diuraphis noxia* (Mordvilko). These two strains were toxic against *D. noxia* when applied with an artificial diet, with 90% mortality with strain B120T and 50% mortality with strain B137 after 2 days.

Another example is the phytopathogenic bacterium *Dickeya dadantii*, the causal agent of soft rot disease. This bacterium was identified by Grenier and colleagues in 2006 [55] as pathogenic to the aphid *A. pisum*. Genome analysis of *D. dadantii* revealed four homologous genes encoding insecticidal proteins of the Cyt family, similar to those found in *B. thuringiensis*. Subsequent research conducted by Costechareyre [56] determined that these four homologous genes are closely related to the genes *cytA*, *cytB*, *cytC*, and *cytD*. Furthermore, the mutation of these genes resulted in a decreased virulence of *D. dadantii* against *A. pisum*, although it remained lethal to the aphid. These findings suggest that Cyt proteins are not the only factors involved in pathogenicity against *A. pisum*, indicating the presence of other virulence factors associated with pathogen–host interactions. Additionally, they show that when the bacteria colonized the gut and stomach of the aphid, they gradually invaded internal and fatty tissue, leading to death by septicemia. Transcriptome analysis showed that *D. dadantii* exhibited a strong defense against antimicrobial peptides due to the expression of a large number of transport proteins and efflux pumps [57,58]. On the other hand, the generalized septicemia induced by Cyt-like entomotoxins (CytC) led to insect death when the bacterial load reached approximately 10^8^ CFU in *A. pisum* [57].

Another study carried out by Renoz and colleagues in 2015 [59] investigated the effects of ingesting the species *Serratia symbiotica* and *S. marcescens* by *A. pisum*. They analyze the expression of immunogenic genes during colonization or bacterial infection. *Serratia symbiotica* is a free-living bacterium that is also found as part of the aphid microbiota. *Serratia symbiotica* did not affect the survival of the aphid but further colonized the gut and did not trigger an immune reaction in the host. On the other hand, *S. marcescens*, a pathogenic bacterium of vertebrates, induced an immune response that killed the aphid in a period of 3 days. Similarly, Baazeem and colleagues [41] report one strain of *S. liquefaciens* (TU-6) with insecticidal activity against the aphids *A. punicae* and *A. illinoisensis*. They evaluated the strain using a cell culture and a concentrated filtrate of the cell culture, which were topically applied to the aphid’s body. They report an LC_50_ of 6.76 × 10^3^ CFU mL^−1^ against *A. illinoisensis* and an LC_50_ of 2.34 × 10^3^ CFU mL^−1^ against *A. punicae*. However, when they applied the concentrated filtrate of the cell culture, they needed high doses to reach 50% mortality in both aphid species.

In 2022, Paliwal and colleagues [52] identified and tested two strains of Enterobacter (*Enterobacter xiangfangensis* and *Enterobacter* sp. strain LA12P41) for their toxicity against various aphid species, including *M. persicae*, *A. fabae*, *B. brassicae*, *M. albifrons*, *N. ribisnigri*, and *A. solani*. Their findings revealed that all tested strains exhibited pathogenicity against every aphid species, resulting in mortality rates ranging from 60% to 80% after 72 h using bacterial concentrations from 10^7^ CFU mL^−1^, whereas at lower concentrations, the mortality was reduced to 20–50%. In addition, Wu and colleagues [60] evaluated the toxicity of bacterial metabolites produced by the symbiotic bacteria of the entomopathogenic nematodes *Photorhabdus luminescens* and *Xenorhabdus bovienii* against *Melanocallis caryaefoliae* (Davis) (black pecan aphids) and *Monellia caryella* (Fitch) (blackmargined aphid). Both bacteria caused significant aphid mortality, with rates ranging from 70% to 90% after five days of exposure to a bacterial concentration of 1.33 × 10^8^ CFU mL^−1^. In addition to the studies mentioned above, Altincicek and colleagues [61] carried out research on the pea aphid *A. pisum* by feeding it an artificial diet contaminated with the *Escherichia coli* strain K-12 MG1645, showing that *E. coli* was able to proliferate within the aphid’s gut, leading to mortality in approximately six days, when aphids ingested a diet containing 10^7^ CFU mL^−1^.

On the other hand, it is worth noting that aphids have a reduced immune system compared to other insects, in which several genes considered crucial for immune function towards bacterial pathogens are absent [26]. This may explain why aphids are susceptible to various bacterial groups of the Enterobacteriaceae family. However, it is well known that aphids need to establish symbiotic interactions with different enterobacteria such as *Rickettsia*, *Wolbachia*, and *Spiroplasma*, which are essential in their defense mechanisms against different pathogens, including Enterobacteriaceae bacteria [27].

### 2.3. Moraxellaceae

This family includes species that colonize the skin and mucous membranes of humans and other animals and can occasionally cause a variety of infections, as well as apparently harmless species occurring in the environment, including water, soil, and foodstuffs. Most species are considered saprophytes of little clinical significance, but a few are important infectious agents. Upon Gram staining, bacterial cells appear as Gram-negative rods, coccobacilli, or diplococci [62].

In 2022, Paliwal and colleagues identified a strain belonging to this family with aphicidal activity [52]. They isolated an *Acinetobacter* sp. and tested its toxicity against various aphid species including *M. persicae*, *A. fabae*, *B. brassicae*, *M. albifrons*, *N. ribisnigri*, and *A. solani*. Their findings revealed that the strain exhibited low pathogenicity against every aphid species, resulting in mortality rates of about 20% after 72 h, using bacterial concentrations ranging from 10^2^ to 10^7^ CFU mL^−1^. This may be due to the fact that some strains of *Acinetobacter* are considered opportunistic pathogens [62], and perhaps the aphids are not among their natural target hosts.

### 2.4. Xanthomonadaceae

The family Xanthomonadaceae are bacillus-shaped Gram-negative bacteria. These organisms are ubiquitous and can be isolated from many environments. Similar to the family Enterobacteriaceae, some Xanthomonadaceae species are pathogenic to humans [63], while others cause a variety of diseases in economically important crops worldwide [64]. Although some are important plant pathogens, there are few reports of insecticidal activity. Hashimoto [51] published the only report of a species of *Xanthomonas* that is an aphid pathogen. He evaluated the aphicidal activity of *Stenotrophomonas maltophilia* (previously known as *Xanthomonas maltophilia*) against the foxglove aphid, *A. solani*, and *A. gossypii*, by spraying 10^7^ CFU of the bacteria on the aphids’ bodies. The bioassays showed that this bacterium causes mortalities of 80% at six days post-inoculation. Additionally, Baazeem and colleagues [41] report one strain of *Stenotrophomonas tumulicola* (T5916-2-1b) with insecticidal activity against the aphids *A. punicae* and *A. illinoisensis*. They evaluated the strains using a cell culture and a concentrated filtrate of the cell culture, which were topically applied to the aphid’s body. They report an LC_50_ of 2.75 × 10 CFU mL^−1^ against *A. illinoisensis* and an LC_50_ of 2.69 × 10 CFU mL^−1^ against *A. punicae* after 48 h of exposure. However, when they applied the concentrated filtrate of the cell culture, they needed high doses (400 to 600 μLmL^−1^) to reach 50% mortality in both aphid species.

The reports mentioned above could suggest that some other members of this family may have insecticidal activity, considering that some genera within this family have a close phytopathogenic relationship with various plants.

### 2.5. Pseudomonadaceae

This family has a wide distribution and is intimately associated with both plants and animals. It is therefore considered to be a ubiquitous taxon. Pseudomonadaceae are non-spore-forming and have a bacillus-like shape, Gram-negative cells, and motility using flagella. Like in other families, some species can be opportunistic pathogens in humans [65]. It is one of the few families from which commercial formulations for agricultural use have been generated and are currently available [66]. For example, Stavrinides and colleagues [53] demonstrated that the phytopathogenic bacterium *Pseudomonas syringae* can use the pea aphid as a vector and as an alternative host. The feeding habit of *A. pisum* facilitates the ingestion of this epiphytic bacterium, which then colonizes the aphid gut and then comes into contact with the phyllosphere of the plant when the aphid excretes honeydew. After several days of ingestion of *P. syringae*, and after having carried out its cycle of growth and dissemination in the phyllosphere of the host plant, the intestinal infection induced in the aphid causes death by bacterial sepsis.

Within the genus *Pseudomonas*, *P. fluorescens* is well known to have antifungal, nematicidal, and plant growth promotion activities, as well as producing different metabolites that have biocidal effects [67,68,69]. The study carried out by Hashimoto [51] showed that *P. fluorescens* was pathogenic to three aphid species—*A. solani*, *A. gossypii*, and *M. persicae*—when sprayed with a bacterial suspension or by ingestion. Mortalities induced by *P. fluorescens* were near 100% at 2 days post-inoculation. Recently, a field evaluation of *P. fluorescens* was carried out in cotton crops against sucking pests. Data obtained from this work revealed that the application of *P. fluorescens* on soil and leaves reduced the abundance of *A. gossypii* by up to 58% after three successive sprays at 15 d intervals. In addition, *P. fluorescens* treatments had minimal side effects on natural enemies compared to chemical treatment, and a significantly higher seed cotton yield was harvested [70]. In 2022, Paliwal and colleagues [52] isolated four strains of Pseudomonas (*Pseudomonas* sp., *P. poae*, *P. rhizosphaerae*, and *P. fluorescens*) and evaluated their toxicity against *M. persicae*, *A. fabae*, *B. brassicae*, *M. albifrons*, *N. ribisnigri*, and *A. solani*. They found that all the strains were pathogenic against all the aphids, resulting in mortality percentages ranging from 80% to 100% after 72 h, using bacterial concentrations ranging from 10^2^ to 10^7^ CFU mL^−1^.

In recent years, different molecules produced by *Pseudomonas* with insecticidal effects against different aphid species, such as biosurfactantes and glycolipids (Table 2), have been identified using advanced spectrometric techniques. In a recent study, Paliwal and colleagues [71] carried out a transcriptomic analysis of the strain *P. fluorescens* PpR24. Their investigation revealed that this particular strain produces a variety of insecticidal toxins and proteases, which play a crucial role in causing mortality in the aphid *M. persicae.* They reported that the strain PpR24 synthesizes toxin complexes (Tc), rearrangement hotspot (Rhs) elements, the protease AprX, and four distinct toxin components (including two TcA-like, one TcB-like, and one TcC-like insecticidal toxins). This comprehensive analysis of host–pathogen interactions sheds new light on the molecular mechanism underlying bacteria-mediated aphid mortality, offering promising insights for its applications as an effective biocontrol agent.

### 2.6. Streptomycetaceae

Streptomycetaceae exhibit mycelial growth and produce spores for survival and propagation; they are obligate aerobic organisms and can be isolated from many environments, but they are primarily abundant in soil [81]. Due to their ability to produce a wide array of bioactive compounds, Streptomycetaceae hold significant biotechnological importance and are extensively studied for potential applications in medicine, agriculture, and industry [82]. For example, the genus *Streptomyces* has been widely utilized in biotechnology processes for the production of antibiotics, fungicides, bactericides, herbicides, insecticides, and acaricides [83]. Avermectins are a group of macrocyclic lactones isolated from *S. avermitilis* that act as agonists of GABA-gate chloride channels [72] and are among the most potent anthelmintic, insecticidal, and acaricidal compounds known. Between the 1950s and 1970s, approximately 60% of new antibiotics were isolated from *Streptomyces* species. Currently, 39% of all bioactive microbial metabolites are produced by the genus *Streptomyces*, and the family Streptomycetaceae contributes 80% of microbial antibiotics [84].

*Streptomyces albus* was isolated as an endophyte from drunken horse grass, *Achnatherum inebrians*, and its insecticidal potential was determined through laboratory bioassays by the nebulization of the culture supernatant against *A. gossypii*. The biocidal effect was observed at 24 h of exposure, producing about 90% mortality [14]. Endophytic actinobacteria isolated from the neem tree, *Azadirachta indica*, showed insecticidal activity on *M. persicae* [85]. In this study, 85 actinobacteria strains were screened for their insecticidal activity on the aphid, and crude extract from eight strains showed mortalities above 60%. Among these eight actinobacteria, strain G30, identified as *S. albidoflavus*, induced 94.6 ± 1.0% of mortality at 48 h after application

### 2.7. Neisseriaceae

The family Neisseriaceae comprises coccobacillary microorganisms that are Gram-negative, flagellated, and non-endospore-forming. Most species within this family are aerobic and chemoorganotrophs [86]. One of the most notable species in this family is *Chromobacterium subtsugae*, described by Martin and colleagues in 2007 [87]. This bacterium is significant because it produces insecticidal factors that are effective against a variety of insect pests. Initial studies assessed its biocidal potential upon ingestion by insects of the orders Coleoptera, Lepidoptera, and Hemiptera. Currently, Marrone Bio Innovations Inc. has registered and developed a commercial formulation based on the *C. subtsugae* strain PRAA4-1T (Grandevo^®^). This biopesticide is recommended for controlling aphids (*A. gossypii*, *M. caryaefoliae*, and *B. brassicae*) [88], various lepidopteran and coleopteran pests (EPA Reg. No.: 84059-17-87865), as well as mites. In addition to insecticidal metabolites, it possesses genes encoding vegetative insecticidal proteins, violacein, and other secondary metabolites that are important for its insecticidal activity [13,19,89].

### 2.8. Brucellaceae

The family Brucellaceae are Gram-negative, non-spore-forming, facultative intracellular pathogens, with some members known for causing brucellosis, a zoonotic disease that primarily affects mammals [90]. Recently, Baazeem and colleagues [41] reported on a strain of *Pseudochrobactrum saccharolyticum* (CCUG 33852) with insecticidal activity against the aphids *A. punicae* and *A. illinoisensis*. They evaluated the strain using cell culture and a concentrated filtrate of the cell culture, which were topically applied to the aphid’s body. They report that the cell culture shows an LC_50_ of 1.26 × 10^7^ CFU mL^−1^ against *A. illinoisensis* and an LC_50_ of 1.62 × 10^6^ CFU mL^−1^ against *A. punicae* after 48 h of exposure. However, when they applied the concentrated filtrate of the cell culture, they needed doses above 500 μLmL^−1^ to reach 50% mortality in both aphid species.

### 2.9. Leuconostocaceae

*Leuconostoc* spp. are important lactic acid bacteria widely employed as starter cultures for fermenting vegetables and dairy products, with potential applications as functional food ingredients [91]. Recently, Hiebert and colleagues [92] isolated the bacterium *Leuconostoc pseudomesenteroides* from *Drosophila suzukii* and evaluated its toxicity against *A. pisum*. Septic infection with *L. pseudomesenteroides* killed all aphids within three days, while oral infection, though less effective, still decreased overall survival after three days by 25% and 30% using concentrations of 4.9 × 10^7^ and 4.9 × 10^8^ CFU mL^−1^; in general, the effect was similar at both doses. When live bacteria or cell-free extracts were administered orally to aphid nymphs, infection with the living bacteria significantly decreased aphid survival. These results confirm the potential of *L. pseudomesenteroides* as a novel biocontrol agent for sustainable pest management.

The diversity of bacteria with entomopathogenic potential against aphids is extensive, as demonstrated by the studies referenced above (Figure 1).

## 3. Possible Reasons for the Limited Toxicity of BPP Against Aphids

Within the group of BPPs, only a limited number have been evaluated in aphids (Table 1), and the majority of them have shown low toxicity against this group of insects. This may be because of feeding habits and characteristics of the aphid gut that do not allow BPPs to efficiently bind or process (Figure 2). For instance, in contrast to the guts of other insect groups, the aphid gut possesses filter chambers and lacks a peritrophic matrix [23]. The filter chamber at the junction between the posterior and anterior midgut, present in the guts of aphids and other sap-sucking Hemiptera [93], is associated with maintaining the osmotic balance between the hemolymph and the gut [94]. Furthermore, it is known that in some sap-feeding insects, this structure does not have an association with aquaporins. This characteristic could explain why some BPPs (e.g., Cry toxins) have lower activity, as the filter chamber keeps them from reaching the appropriate region to interact with the receptors. On the other hand, hemipterans lack a peritrophic membrane, which in other insects is important for binding Cry toxins and limiting their movement into the gut to interact with receptors [95,96,97]. The absence of this structure may result in faster excretion of BPPs, as demonstrated by Brand and colleagues [98] in *Lygus hesperus* (Knight) (Hemiptera: Miridae), where Cry1Ac was eliminated in feces. Other explanations for this lower toxicity could be inappropriate gut enzymes and pH levels. In lepidopterans, for example, the majority of Cry and Cyt proteins are activated by serine proteases under alkaline conditions, with pH ranging from 7.5 to 12 [31]. In contrast, the pH of the gut membrane and lumen of some aphids like *A. pisum* ranges from 4.4 to 7 [99], which may result in the incomplete hydrolysis of Cry1Ac and Cry3Aa by cysteine proteases [20]. Similarly, the gut contents of *A. pisum* failed to activate Cry4Aa [9]. The lack of appropriate activation of these proteins could be an explanation for why Cry proteins are not effective against aphids.

## 4. Mode of Action of Bacterial BPPs in Aphids

Although functional receptors for BPPs have not been identified in sap-sucking pests, some studies have investigated this association. For example, Zhao and colleagues [100] identified possible receptors for the Cry41 toxin in the *M. persicae* gut, including cathepsin B, calcium-transporting ATPase, and *Buchnera*-derived ATP-dependent 6-phosphofructokinase (PFKA). They showed that binding to cathepsin B increased Cry41-related toxin activity, which may result in an acceleration of the apoptosis of aphid cells. Additionally, Jin and colleagues [22] reported that the binding of Cry41 to the symbiont protein *Buchnera*-derived PFKA decreases the number of *Buchnera* symbionts, suggesting that the toxin may kill *M. persicae* by inhibiting the activity of *Buchnera*-derived PFKA.

It is well documented that aphids supply non-essential amino acids to their symbiotic organisms, and as a result, the symbiont offers essential amino acids or vital components in the host’s biosynthetic pathway [101]. However, some nutrients provided by the symbiont to aphids need to be synthesized in the metabolic pathways of *Buchnera*, involving enzymes that are coded by the host [102], suggesting that aphids cannot survive without *Buchnera*. In this sense, Jin and colleagues [22] proposed that after the ingestion of the Cry41-related toxin by *M. persicae*, the toxin enters the mycetocyte and *Buchnera* via an unknown receptor. It then interacts with ATP-dependent 6-phosphofructokinase (PFKA) to reduce its activity in *Buchnera*, leading to the accumulation of upstream fructose-6-phosphate that impairs cell viability, eventually resulting in a significant decrease in the number of *Buchnera*. Another possible entry point is that the Cry41-related toxin may bind to another unknown receptor, and then the toxin interacts with cathepsin B to enhance its activity, leading to the accelerated apoptosis of aphid cells. These two mechanisms result in the death of *M. persicae* nymphs (Figure 2).

## 5. Conclusions and Future Perspectives

The food needs of the human population gave rise to the beginning of agricultural production over large extents of land; this brought with it populations of insect pests that can increase drastically. The indiscriminate application of chemical insecticides to counteract these pests can cause resistance to insecticides and eliminate natural enemies and other non-target organisms, causing an irreversible loss of fauna. Since the last century, researchers in the area of biological control have been given the task of searching for entomopathogenic bacteria that can be used as an environmentally benign alternative to insecticides, thus helping to reduce the use of chemical agents. The molecular identification of these organisms by applying sequencing techniques has allowed knowledge within the scientific community to advance at great leaps over the past few decades, leading to the discovery of new bacterial species with particular characteristics that allow them to be applied in the management of pests and diseases. The elucidation of the mode of action of the toxins and metabolites that are responsible for toxicity may enhance the discovery of new species with potential insecticidal applications. In the coming years, it is crucial to implement novel biotechnological and bioinformatic approaches aimed at developing alternative pest management solutions for crops. These strategies could involve harnessing microbial agents and their toxins for crop genetic enhancement through biotechnological methods. Remarkably, only around 5% of known bacteria have been characterized, with many unable to be cultured and countless others yet to be discovered. Nature serves as an expansive laboratory, offering a vast array of molecules with diverse potential applications, ranging from therapeutic uses to agricultural pest control. As our understanding of bacteria and their potential applications expands in the decades ahead, it will open up a plethora of opportunities and alternatives for managing agricultural pests and vectors.

## Figures and Tables

**Figure 1 plants-14-00943-f001:**
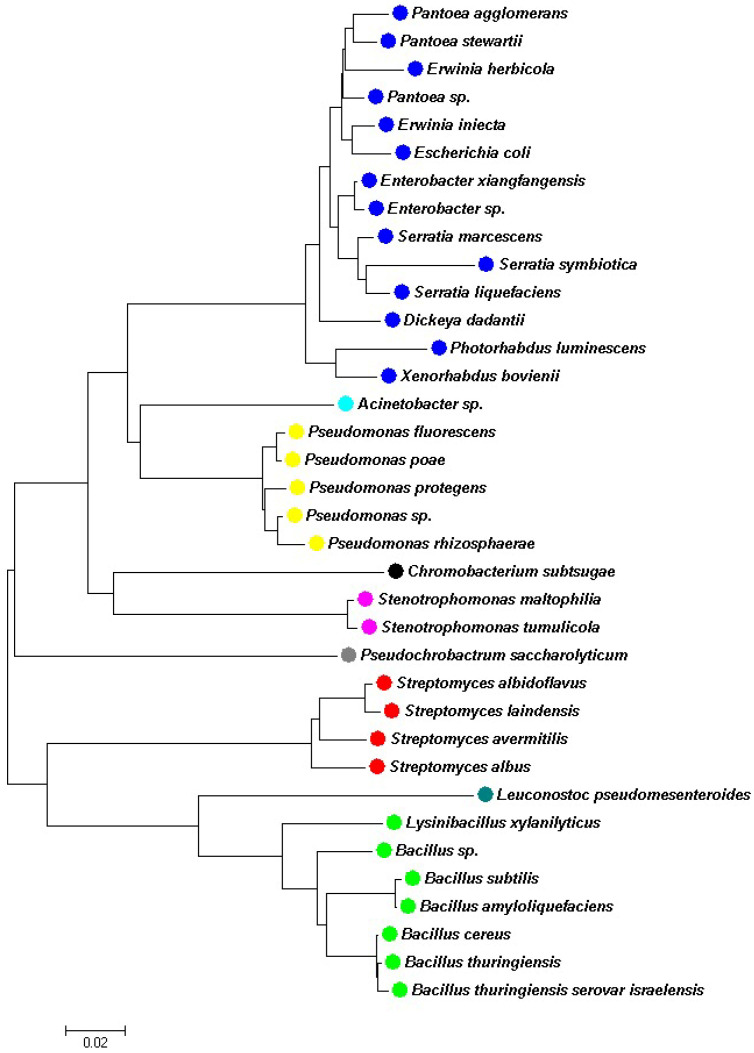
The phylogenetic tree illustrates the diversity of bacteria with entomopathogenic activity against aphids. Families: Enterobacteriaceae, Moraxellaceae, Pseudomonadaceae, Neisseriaceae, Xanthomonadaceae, Brucellaceae, Streptomycetaceae, Leuconostocaceae, and Bacillaceae.

**Figure 2 plants-14-00943-f002:**
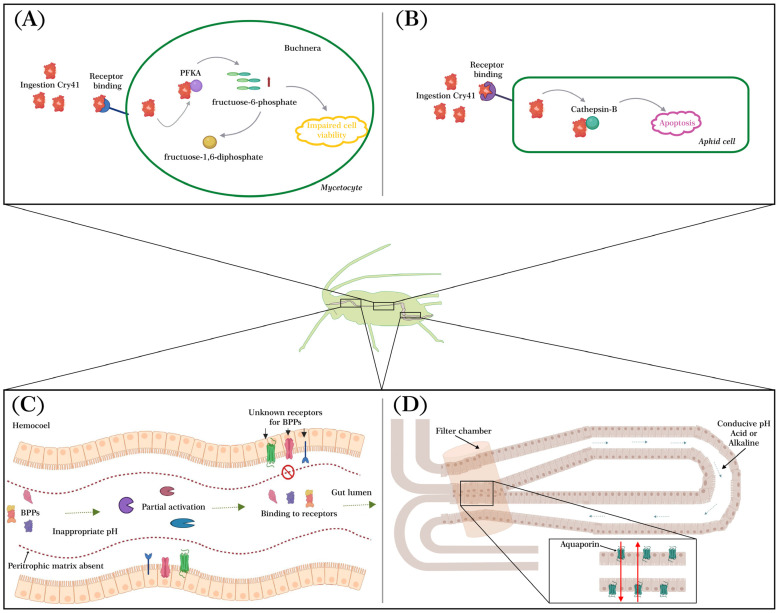
The figure illustrates the modes of action of *Bacillus thuringiensis* bacterial pesticidal proteins (BPPs), specifically Cry41-related toxins (**A**,**B**) and characteristics of the aphid gut that hinder the efficient bonding of BPPs (**C**,**D**). (**A**) Upon ingestion by aphids, a portion of the Cry41-related toxin gains access to the mycetocyte and *Buchnera* via an unidentified receptor (receptor 1). Subsequently, it interacts with ATP-dependent 6-phosphofructokinase (PFKA), reducing its activity within *Buchnera*. This enzymatic interaction results in the accumulation of fructose-6-phosphate upstream, leading to impaired cell viability and a subsequent decrease in the population of *Buchnera*. (**B**) Cry41-related toxin may potentially infiltrate aphids through another unidentified receptor (receptor 2). Upon entry, it interacts with cathepsin B, enhancing Cry41-related toxin activity. This heightened enzymatic activity accelerates apoptosis within aphid cells, ultimately resulting in the demise of aphid nymphs. (**C**) Possible reasons for the limited toxicity of BPPs against aphids. (**D**) The diagram depicts a filter chamber located at the junction between the posterior and anterior midgut, with red arrows indicating the direction of water movement. The figure was created and adapted using information from [22,24].

**Table 1 plants-14-00943-t001:** Bacterial pesticidal proteins (BPPs) reported to have insecticidal activity against aphids.

Target Aphid	Protein	% Mortality	ng/mL (Time)	Origin	Reference
Potato aphid *Macrosiphum euphorbiae*	Mixture CryI; CryIA(a), CryIA(b), CryIC and CryIF CryIIA CryIIIA CryIVD	100 ± 0 93 ± 0 98 ± 4 93 ± 10	100 each Cry (4 days) 200 (4 days) 360 (4 days) 350 (4 days)	Recombinant strains of Bt	[25]
Pea aphid *Acyrthosiphon pisum*	CryIAb Cry3A Cry4A and Cry4B Cry11A CytIA	35 60 100 100 Growth inhibition	500 (5 days) 500 (6 days) 500 (4 days) 500 (3 days) 125 (7 days)	Recombinant strains Bt subsp. *israelensis*	[9]
Cotton aphid *Aphis gossypii*	VipIAcI and Vip2Ae3	LC_50_	0.0875 (NR days)	*Bacillus cereus*	[42]
Pea aphid *Acyrthosiphon pisum*	CryIAc Cry3Aa	71 71	500 (7 days)	*Bacillus thuringiensis*	[20]
Cotton aphid *Aphis gossypii*	VipIAe and Vip2Ae	LC_50_	0.576 (2 days)	*Bacillus thuringiensis*	[43]
Pea aphid *Acyrthosiphon pisum*	Cyt2Aa CGAL1 CGAL3 CGAL4 CGSL1 CGSL4	LC_50_ LC_50_ LC_50_ LC_50_ LC_50_ LC_50_	150 ± 0.00 19.71 ± 5.74 9.55 ± 2.54 11.92 ± 1.99 28.74 ± 2.92 15.13 ± 0.23	Recombinant proteins	[23]
Green peach aphid *Myzus persicae*	Cyt2Aa CGAL1 CGAL3 CGAL4 CGSL1 CGSL4	LC_50_ LC_50_ LC_50_ LC_50_ LC_50_ LC_50_	150 ± 0.00 58.04 ± 2.08 42.68 ± 0.49 92.75 ± 2.54 ND ND	*Bacillus thuringiensis* Recombinant proteins	[23]
Green peach aphid *Myzus persicae*	Cry-Related	LC_50_	32.7 (3 days)	*Bacillus thuringiensis*	[10]
Pea aphid *Acyrthosiphon pisum*	CytIA CytC CytB CytA	TL_50_ TL_50_ TL_50_ TL_50_	1000 (3.24 days) 1000 (10.1 days) 500 (5.1 days) 1000 (2.28 days)	*Dickeya* *dadantii*	[44]
Cotton aphid *Aphis gossypii*	Cry1Ah Cry2Ab	No mortality	>1000	*Bacillus thuringiensis*	[21]
Cabbage aphid *Brevicoryne brassicae* Green peach aphid *Myzus persicae*	Cry1Ac Cry1F Cry1Ac and Cry1F Cry1Ac Cry1F Cry1Ac and Cry1F	Decreased the net population growth rate Decreased the net population growth rate	20 (3 days) 20 (3 days) 20 (3 days) 20 (3 days) 20 (3 days) 20 (3 days)	*Bacillus* sp.	[45]
Pea aphid *Acyrthosiphon pisum*	Cry4Aa (trypsin activated) Cry4Aa 2A	63.3 ± 24.5 51.1 ± 2.2	120 (2 days) 120 (2 days)	*Bacillus thuringiensis* Modified toxin	[46]
Green peach aphid *Myzus persicae*	Cry1Cb2	LC_50_	6.58 (3 days)	*Bacillus* sp.	[38]

NR = Not reported; ND = Not determined.

**Table 2 plants-14-00943-t002:** Surfactants and other bacteria-derived molecules with aphicidal properties.

**Target Aphid**	**Molecule**	**% Mortality, or LC_50-90_**	**μg/mL (Time)**	**Origin**	**Reference**
Pea aphid *Acyrthosiphon pisum*	Avermectin B_1_	LC_90_	0.4 ppm (72–96 h)	*Streptomyces avermitilis*	[72,73]
Cotton aphid *Aphis gossypii* Black bean aphid *Aphis fabae*	Avermectin B_1_	50%	450 ppm (NM)	*Streptomyces avermitilis*	[74]
Cotton aphid *Aphis gossypii*	Viscosin	90–99%	200 ppm (6 days)	*Pseudomonas fluorescens*	[51]
Green peach aphid *Myzus persicae*	Dirhamnolipid	100%	100 μg/mL (24 h)	*Pseudomonas* sp.	[75]
Green peach aphid *Myzus persicae*	Orfamide A	LC_50_	34.5 μg/mL (24 h)	*Pseudomonas protegens*	[76]
Green peach aphid *Myzus persicae*	Surfactin	LC_50_	35.82 μg/mL (24 h)	*Bacillus amyloliquefaciens*	[42]
Mustard aphid *Lipaphis erysimi*	Peptide	100%	700 μg/mL (48 h)	*Streptomyces laindensis*	[77]
Green peach aphid *Myzus persicae*	Surfactin isomers	LC_50_	20.4 μg/mL 22.2 μg/mL 54.5 μg/mL (24 h)	*Bacillus subtilis*	[78]
Green peach aphid *Myzus persicae*	Xantholysins A and B	LC_50_	13.4 μg/mL 24.6 μg/mL (24 h)	*Pseudomonas* sp.	[79]
Green peach aphid *Myzus persicae*	Kurstakins, Surfactins, Iturins, Fngycins	No mortality	ND	*Bacillus* *amyloliquefaciens strains* CBMDDrag3, PGPBacCA2	[39]
Greenbug Aphid *Schizaphis graminum*	Surfactin Iturin lipopeptides	100% Using a mix of both strains	112 μg/mL (5 days)	*Bacillus subtilis* strains 26D and 11VM	[80,81]

## Data Availability

The original contributions presented in the study are included in the article; further inquiries can be directed to the corresponding author.

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
