# Peer review of "Entomopathogenic Bacteria Species and Toxins Targeting Aphids (Hemiptera: Aphididae): A Review"

_plants, 2025, doi:10.3390/plants14060943_

Round 1
Reviewer 1 Report
Comments and Suggestions for Authors
This is a very inclusive review regarding entomopathogenic bacterial diversity associated with Aphids, and toxins part, although understudied, was also well reviewed. Some suggestions for the authors to consider:
1) Add a phylogeny to show the bacteria diversity.
2) Please briefly address the diversity of beneficial bacteria associated with Aphids, and their roles in Aphid defense against entomopahtogenic bacteria.
3) Probably briefly address symbiotic fungi associated with Aphids and their possible roles.
4) Please review microbiome studies on Aphids, it is not expected that such a review missed most recent microbiome studies, which are very important for understanding bacteria associated with the insects with a advanced technology. (such as Ma, Yj., He, Hp., Zhao, Hm. et al. Microbiome diversity of cotton aphids (Aphis gossypii) is associated with host alternation. Sci Rep 11, 5260 (2021). https://doi.org/10.1038/s41598-021-83675-2
Kaszyca-Taszakowska N, Depa Ł. Microbiome of the Aphid Genus Dysaphis Börner (Hemiptera: Aphidinae) and Its Relation to Ant Attendance. Insects. 2022 Nov 26;13(12):1089. doi: 10.3390/insects13121089. PMID: 36554999; PMCID: PMC9781600.
Wolfgang, A., Tack, A.J.M., Berg, G. et al. Reciprocal influence of soil, phyllosphere, and aphid microbiomes. Environmental Microbiome 18, 63 (2023). https://doi.org/10.1186/s40793-023-00515-8
Chen Xiaoyulong , Yang Hong , Cernava Tomislav 2021. Microbiome Structure of the Aphid Myzus persicae (Sulzer) Is Shaped by Different Solanaceae Plant Diets JOURNAL=Frontiers in Microbiology)
Author Response
Add a phylogeny to show the bacteria diversity.
In response to the reviewer’s suggestion, the phylogenetic tree was added to show the diversity of the bacteria.
2) Please briefly address the diversity of beneficial bacteria associated with Aphids, and their roles in Aphid defense against entomopahtogenic bacteria.
There is no information about bacterial symbionts associated with aphids and their protective effects against entomopathogenic bacteria. Although the mentioned topic is relevant, it is not the objective of this review. We are focused on bacteria, toxins, and other metabolites with aphicidal effects, and the possible mechanisms by which these toxins affect aphids.
3) Probably briefly address symbiotic fungi associated with Aphids and their possible roles.
There is information about bacterial symbionts associated with aphids and their protective effects against entomopathogenic fungi, but there is no information regarding symbiotic fungi associated with aphids and their protective role.
4) Please review microbiome studies on Aphids, it is not expected that such a review missed most recent microbiome studies, which are very important for understanding bacteria associated with the insects with a advanced technology.
Ma, Yj., He, Hp., Zhao, Hm. et al. Microbiome diversity of cotton aphids (Aphis gossypii) is associated with host alternation. Sci Rep 11, 5260 (2021). https://doi.org/10.1038/s41598-021-83675-2
This article demonstrates that the diversity of the microbiome in aphids depends on the host plant.
Kaszyca-Taszakowska N, Depa Ł. Microbiome of the Aphid Genus Dysaphis Börner (Hemiptera: Aphidinae) and Its Relation to Ant Attendance. Insects. 2022 Nov 26;13(12):1089. doi: 10.3390/insects13121089. PMID: 36554999; PMCID: PMC9781600.
This article shows that aphid colonies attended by ants were richer in genera than unattended aphid colonies.
Wolfgang, A., Tack, A.J.M., Berg, G. et al. Reciprocal influence of soil, phyllosphere, and aphid microbiomes. Environmental Microbiome 18, 63 (2023). https://doi.org/10.1186/s40793-023-00515-8
This study examined the effect of soil on plant and aphid microbiomes, and the reciprocal effect of aphid herbivory on the plant and soil microbiomes.
Chen Xiaoyulong , Yang Hong , Cernava Tomislav 2021. Microbiome Structure of the Aphid Myzus persicae (Sulzer) Is Shaped by Different Solanaceae Plant Diets (Frontiers in Microbiology)
This article shows that aphids transferred to different plants from the Solanaceae family resulted in a substantial decrease in the abundance of the primary symbiont. These results provide strong evidence that the aphid microbiome responds to different plant diets.
The studies mentioned above are interesting; however, these works are outside the scope of this review.
Reviewer 2 Report
Comments and Suggestions for Authors
This manuscript is a comprehensive review of various bacteria and their metabolites investigated as potential biocontrol agents against aphids. The review is organized systematically by bacteria families which is reasonable and accompanied by two sections on mode of action and limited toxicity of bacterial pesticidal proteins in aphids. The manuscript is well written, nevertheless, some revision is required before it could be accepted for publication. See specific comments below (numbers indicate lines in ms).
Title: it is fine and well describes the content of the review
Abstract: few more information would be nice to include, e.g. which group of bacteria is the most promising/effective, if there were also any field trials or commercial biopesticide against aphids etc.
Keywords: few the most important bacteria species or toxins can be named instead of words "aphids" and "toxins" which are already in title
Further text in ms: write insect species in full with author name only at the first occurrence, then use abbreviated form (e.g. 110, 140, 183-185 etc.).
34-35 this sentence would deserve reference(s)
38 full stop is missing after reference after 1,2,3]
49-55 these two sentences might be rewritten for better reading
73-75 information on commercial application of entomopathogenic fungi seems to be a bit out of context so I suggest to remove it
At the end of Introduction aim of the present review should be mentioned with information what reader could expect, e.g. how is the review organized etc.
126 Since this review is on aphids and no higher classification is given in other aphid species, I suggest to delete "(Homoptera: Aphididae)". In case of other pests, the authors can consider adding order: family when species name is mentioned for the first time.
189 it seems some word is missing after "significant"
Note to B. thuringiensis and Cry toxins: there were transgenic (Bt) crops developed and their effects also on aphids were studied. Perhaps something on this could be mentioned in this section, too.
215-216 it is better to write ranges from lower to higher values
225 should read "noxia"
265-266 it would be good to mention that both are symbiotic bacteria found in entomopathogenic nematodes
295 Acinetobacter should be in italics
305 They > He
346 reference number [52] should be moved after "colleagues"
380 antihelmintic
431 concentrations should be swapped (lower first)
453 add (Knight) (Heteroptera: Miridae) after hesperus
Figure 1: this must be substantially improved because it has very low quality/resolution and text is is too small and impossible to read even when magnified in PDF viewer.
481 for consistency with previous text, write only BPPs because this abbreviation has been already defined earlier
Author Response
Abstract: few more information would be nice to include, e.g. which group of bacteria is the most promising/effective, if there were also any field trials or commercial biopesticide against aphids etc.
Additional information was included in the abstract to make it more engaging and appealing to readers.
Keywords: few the most important bacteria species or toxins can be named instead of words "aphids" and "toxins" which are already in title.
Keywords were improved according to the reviewer’s comment.
Further text in ms: write insect species in full with author name only at the first occurrence, then use abbreviated form (e.g. 110, 140, 183-185 etc.).
The full scientific names of insect species, including the authors' names, were provided only at their first mention, following the reviewer's recommendation.
34-36 this sentence would deserve reference(s).
A reference was added to this sentence, following the reviewer's recommendation.
38 full stop is missing after reference after 1,2,3].
A full stop was added, following the reviewer's recommendation.
49-55 these two sentences might be rewritten for better reading
These two sentences have been rewritten for improved readability.
73-75 information on commercial application of entomopathogenic fungi seems to be a bit out of context so I suggest to remove it.
The sentence was removed following the reviewer's comment.
At the end of Introduction aim of the present review should be mentioned with information what reader could expect, e.g. how is the review organized etc.
Line 91-94: We included an objective, mentioning to the reader how the review is organized.
128: Since this review is on aphids and no higher classification is given in other aphid species, I suggest to delete "(Homoptera: Aphididae)".
We deleted "(Homoptera: Aphididae)".
191: it seems some word is missing after "significant".
The sentence was modified to be clearer.
Note to B. thuringiensis and Cry toxins: there were transgenic (Bt) crops developed and their effects also on aphids were studied. Perhaps something on this could be mentioned in this section, too.
Transgenic crops were developed to target Lepidopteran pests such as Spodoptera frugiperda and Heliothis zea, among others. The toxicity of these Bt crops was evaluated, focusing on the effects they may have on other insects, including aphids. The results demonstrated null or insignificant toxicity. For this reason, we consider these studies outside the scope of our work, as these crops were never designed to target aphids.
222-223 it is better to write ranges from lower to higher values
The ranges were ordered from lower to higher values.
232 should read "noxia"
The word was corrected.
271-275 it would be good to mention that both are symbiotic bacteria found in entomopathogenic nematodes.
The phrase “symbiotic bacteria of the entomopathogenic nematodes” was added according to the reviewer's comment.
303 Acinetobacter should be in italics
Acinetobacter was italicized.
313 They > He
The pronoun 'He' was added to the sentence.
354 reference number [52] should be moved after "colleagues"
The reference was moved to after “colleagues”
388 antihelmintic
We found that “anthelmintics” is the most commonly accepted and widely used term.
438 concentrations should be swapped (lower first)
The concentrations were swapped.
460 add (Knight) (Heteroptera: Miridae) after Hesperus
The words “(Knight) (Hemiptera: Miridae)” were added.
Figure 1: this must be substantially improved because it has very low quality/resolution and text is is too small and impossible to read even when magnified in PDF viewer.
Figure 1 was improved based on the reviewer's comments.
488 for consistency with previous text, write only BPPs because this abbreviation has been already defined earlier.
Only the acronym “BPPs” was used in the title section.
Round 2
Reviewer 1 Report
Comments and Suggestions for Authors
It is a bit unfortunate that the authors dedicated time and effort to answering my questions regarding recent achievements in microbiomes associated with aphids but did not briefly address how these advancements are relevant to the topic in the text. A review would benefit from including some future perspectives!
Reviewer 2 Report
Comments and Suggestions for Authors
The authors addressed all my suggestions and comments. The revised version has been improved and I do not have any more recommendations for additional revision.